# Diagnostic Management of Gastroenteropancreatic Neuroendocrine Neoplasms: Technique Optimization and Tips and Tricks for Radiologists

Fabio Pellegrino [1,*], Vincenza Granata [2], Roberta Fusco [3], Francesca Grassi [4,5], Salvatore Tafuto [6], Luca Perrucci [7], Giulia Tralli [8] and Mariano Scaglione [9]

1   Radiology Division, S. Bonifacio Hospital, 37047 Verona, Italy
2   Division of Radiology, Istituto Nazionale Tumori IRCCS Fondazione Pascale—IRCCS di Napoli, 80131 Naples, Italy
3   Medical Oncology Division, Igea SpA, 80013 Naples, Italy
4   Italian Society of Medical and Interventional Radiology (SIRM), SIRM Foundation, Via della Signora 2, 20122 Milan, Italy
5   Division of Radiology, Università degli Studi della Campania Luigi Vanvitelli, 80127 Naples, Italy
6   S.C. Sarcomi e Tumori Rari, Istituto Nazionale Tumori, IRCCS, Fondazione "G. Pascale", 80131 Naples, Italy
7   Ferrara Department of Interventional and Diagnostic Radiology, Ospedale di Lagosanto, Azienda AUSL, 44023 Ferrara, Italy
8   Department of Radiology, Ospedale Santa Maria della Misericordia, 45100 Rovigo, Italy
9   Department of Medical, Surgical and Experimental Sciences, University of Sassari, 07100 Sassari, Italy
*   Correspondence: fabio.pellegrino@aulss9.veneto.it

**Abstract:** Gastroenteropancreatic neuroendocrine neoplasms (GEP-NENs) comprise a heterogeneous group of neoplasms, which derive from cells of the diffuse neuroendocrine system that specializes in producing hormones and neuropeptides and arise in most cases sporadically and, to a lesser extent, in the context of complex genetic syndromes. Furthermore, they are primarily nonfunctioning, while, in the case of insulinomas, gastrinomas, glucagonomas, vipomas, and somatostatinomas, they produce hormones responsible for clinical syndromes. The GEP-NEN tumor grade and cell differentiation may result in different clinical behaviors and prognoses, with grade one (G1) and grade two (G2) neuroendocrine tumors showing a more favorable outcome than grade three (G3) NET and neuroendocrine carcinoma. Two critical issues should be considered in the NEN diagnostic workup: first, the need to identify the presence of the tumor, and, second, to define the primary site and evaluate regional and distant metastases. Indeed, the primary site, stage, grade, and function are prognostic factors that the radiologist should evaluate to guide prognosis and management. The correct diagnostic management of the patient includes a combination of morphological and functional evaluations. Concerning morphological evaluations, according to the consensus guidelines of the European Neuroendocrine Tumor Society (ENETS), computed tomography (CT) with a contrast medium is recommended. Contrast-enhanced magnetic resonance imaging (MRI), including diffusion-weighted imaging (DWI), is usually indicated for use to evaluate the liver, pancreas, brain, and bones. Ultrasonography (US) is often helpful in the initial diagnosis of liver metastases, and contrast-enhanced ultrasound (CEUS) can solve problems in characterizing the liver, as this tool can guide the biopsy of liver lesions. In addition, intraoperative ultrasound is an effective tool during surgical procedures. Positron emission tomography (PET-CT) with FDG for nonfunctioning lesions and somatostatin analogs for functional lesions are very useful for identifying and evaluating metabolic receptors. The detection of heterogeneity in somatostatin receptor (SSTR) expression is also crucial for treatment decision making. In this narrative review, we have described the role of morphological and functional imaging tools in the assessment of GEP-NENs according to current major guidelines.

**Keywords:** gastroenteropancreatic; neuroendocrine; neoplasms; diagnosis; radiology; somatostatin receptor imaging; PET; computed tomography; magnetic resonance; ultrasound

## 1. Introduction

Gastroenteropancreatic neuroendocrine neoplasms (GEP-NENs) account for approximately 1.5% of all gastrointestinal and pancreatic malignancies [1]. The word "neuroendocrine" derives from the similarity of these cells with the neural crest for the expression of, for example, synaptophysin, neurospecific enolase and chromogranin proteins [2,3]. Although it is known that the incidence is low, the growth is slow, and the prevalence is high, in recent decades, there has been an increase in their incidence rate, also due to the improvement of radiological, nuclear medical, and endoscopic imaging techniques [4,5]. Most of these neoplasms arise sporadically, although some cases occur in complex genetic syndromes. In 60–80% of cases, they are nonfunctional, while in 20–30% of cases, these lesions (e.g., insulinomas, gastrinomas, glucagonomas, vipomas, and somatostatinoma) produce hormones responsible for clinical syndromes [6]. With regard to clinical symptoms, in patients with insulinomas, excessive insulin secretion causes hypoglycemia and hypokalemia; pancreatic polypeptide-secreting tumor (Ppomas) patients have weight loss, abdominal pain, and jaundice. In vipomas lesions, we found watery diarrhea, achlorhydria, and hypokalemia. The 4 Ds, diabetes, dermatitis (necrolytic migratory erythema), deep vein thrombosis, and depression, are typical of glucagonomas, while, for somatostatinomas, there are cholelithiasis, hyperglycemia, and steatorrhea.

Regarding enteric lesions, carcinoid syndrome with watery diarrhea, hypotension, bronchospasm, flushing, and right-sided heart disease is due to serotonin hypersecretion. Since serotonin from small bowel lesions is drained through the portal vein in the liver, where monoamine oxidases inactivate it, this syndrome appears if hepatic metabolism is avoided, for example, if the liver or other distant metastases occur (Table 1).

**Table 1.** Classification of GEP-NETs based on clinical syndromes.

| | |
|---|---|
| Hypoglycemia and hypokalemia | Insulinomas |
| Weight loss, abdominal pain, and jaundice | Ppomas |
| Watery diarrhea, achlorhydria, and hypokalemia | Vipomas |
| Diabetes, dermatitis (necrolytic migratory erythema), deep vein thrombosis, and depression | Glucagonomas |
| Cholelithiasis, hyperglycemia, and steatorrhea | Somatostatinomas |
| Carcinoid syndrome with watery diarrhea, hypotension, bronchospasm, flushing, and right-sided heart disease | Liver metastasis from small bowel lesions |

In this context, the diagnosis may be easy for functioning lesions, while nonspecific symptoms often lead to a delayed diagnosis for indolent lesions. Therefore, an accurate diagnosis requires a multidisciplinary approach, including biochemical analyses and multiple imaging techniques [7–11]. The clinical course of these biologically diverse tumors is highly varied, ranging from indolent to aggressive [12], with histological differentiation and grading being the most significant factors for prognosis and treatment planning [13]. In the prognosis and management of GEP-NENs, the Ki-67 labelling index and mitotic count are critical indicators of the state of tumor proliferation [14,15]. The WHO classifications of NENs have been updated from previous classifications differentiating neuroendocrine tumors (NETs) and poorly differentiated neuroendocrine carcinoma (NECs), according to the Ki-67 index and mitotic count [16–22] (Table 2).

**Table 2.** The World Health Organization (WHO) 2019 classification for neuroendocrine neoplasms of the gastrointestinal tract and hepatopancreatobiliary organs.

| Terminology | Differentiation | Grade | Mitotic Rate * | Ki-67 Index |
|---|---|---|---|---|
| NET, G1 | Well differentiated | Low | <2 | <3% |
| NET, G2 | Well differentiated | Intermediate | 2–20 | 3–20% |
| NET, G3 | Well differentiated | High | >20 | >20% |
| NEC, small-cell type | Poorly differentiated | High | >20 | >20% |
| NEC, large-cell type | | | >20 | >20% |
| MiNEN | Well or poorly differentiated | Variable | Variable | Variable |

* Mitoses/10 hpf.

In this type of tumor, discrepant tumor grades in the same patient can be determined through intratumoral heterogeneity with different Ki-67 indices in different portions of the same mass or between various tumor sites. Therefore, the aggressiveness of the patient's tumor burden could be misinterpreted with small biopsy specimens in a single location of the lesion [23]. Furthermore, histological samples at a given time do not always represent the grade of the tumor during the entire course of the disease, as the behavior of neuroendocrine tumors is complicated by the fact that these tumors change from low to high grade in time with a more aggressive course of the disease [24]. Two critical issues should be considered in the diagnostic workup of NENs: first, the need to identify tumor presence, and second, to define the primary site and assess regional and distant metastases. In fact, the primary site, stage, grade, and functionality are prognostic factors that the radiologist should assess to guide prognosis and management. Proper diagnostic patient management includes the combination of morphological and functional assessments. Regarding morphological evaluations, according to the consensus guidelines of the European Neuroendocrine Tumor Society (ENETS), contrast-enhanced computed tomography (CT) of the neck–thorax–abdomen and pelvis, including a three-phase examination of the liver, is recommended for the diagnosis, staging, treatment response assessment, and surveillance of primary neuroendocrine tumors. Contrast-enhanced magnetic resonance imaging (MRI), including diffusion-weighted imaging (DWI), is indicated for use to evaluate the liver, pancreas, brain, and bones. Ultrasonography (US) is often helpful in the initial diagnosis of liver metastases as this tool can guide liver lesion biopsies for undefined primary tumors, and contrast-enhanced ultrasound (CEUS) may be used for problem-solving in characterizing liver lesions that remain ambiguous on CT/MRI. In addition, intraoperative US, with or without a contrast medium, represents an efficient tool during surgical procedures, favoring lesion detection. Additionally, it is reported that endoscopic US is the most sensitive modality for diagnosing pancreatic NENs (PNENs), also allowing for lesion biopsies [25]. Regarding functional assessments, somatostatin receptor imaging or 18FDG-PET/CT represent essential tools during the staging, response evaluation, and surveillance [25,26]. In this article, we have reported on the state-of-the-art imaging techniques for GEP-NENs, focusing on the role and optimization of morphological and functional techniques in managing these patients. Our aim is to assist the imaging specialist in choosing and using imaging techniques based on the clinical scenario of the GEP-NEN patients.

## 2. Diagnosis, Staging, and Risk Assessment

A histological diagnosis is mandatory in all patients, and can be carried out on surgical specimens or metastasis biopsies. However, some factors can reduce the adequacy of sampling related to the characteristics of the lesion, such as localization in the pancreatic body or the tail, rather than in the head, or a small size, rich stromal fibrosis, or cystic and necrotic components, but also operative factors, such as the initial experience of

endosonographers and cytopathologists [27]. Disease stage and tumor grade are the two major independent prognostic features that should be evaluated. Therefore, the role of imaging is to detect the primary site to obtain the stage of the disease and to identify the proper patient treatment [28]. In this scenario, functional and morphological evaluations may be used complementarily to overcome the limits of one with the other and vice versa (Table 3). PET/CT or SPECT/CT imaging using adequate tracers allows to accurately evaluate the extension of the disease, both in the initial staging and in the follow-up, and can also identify a primary occult tumor, a task that is sometimes challenging but helpful in optimizing the therapeutic strategy, especially in patients with metastasis disease. Furthermore, functional imaging allows for the noninvasive characterization of the functional status and tumor heterogeneity based on the analysis of the uptake intensity of target-specific radiotracers [29]. Moreover, functional imaging can offer an improved prognostic stratification and refinement of therapeutic strategies, enabling a personalized theranostic approach to managing GEP NETs. Contrast-enhanced CT and MRI supply detailed anatomical information about the primary tumor site and identify regional and distant metastases, information required for optimized surgery, treatment selection, and the identification of persistent or recurrent disease. Finally, endoscopic procedures, such as endoscopic ultrasound, have been used with great success to identify lesions that might otherwise have escaped imaging modalities [30].

**Table 3.** Comparison of imaging techniques for GEP-NEN diagnosis.

| | Advantages | Disadvantages | Use |
|---|---|---|---|
| US | -no ionizing radiations<br>-widely available<br>-inexpensive | -interoperator variability<br>-not recommended for other sites in the gastrointestinal tract<br>-low sensitivity in pancreatic lesions | -initial diagnosis of liver metastases<br>-surveillance in some patients with liver metastases<br>-liver lesion characterization with contrast-enhanced ultrasound (CEUS)<br>-tool to guide liver lesion biopsies<br>-intraoperative use for lesions detection |
| EUS | -possibility of fine-needle biopsy of the lesion with Ki-67 evaluation<br>-increases the overall PNEN detection rate after a CT scan | -inadequacy of sampling due to site (pancreatic body or tail), small size, rich stromal fibrosis, or cystic and necrotic components<br>-inexperience of endosonologists and cytopathologists | -detection of small pancreatic NENs<br>-histological diagnosis<br>-intratumoral vascularity with contrast-enhanced endoscopic ultrasound |
| CT contrast enhanced | -good sensitivity and specificity<br>-widely available | -ionizing radiation | -diagnosis, staging, treatment response assessment, and surveillance of primary neuroendocrine tumors<br>-lung lesions assessment |
| MRI contrast enhanced | -no ionizing radiation<br>-ability to characterize lesions using pre and postcontrast sequences and DWI<br>-cholangiopancreatography sequences to assess the relationship between the lesion and the pancreatic duct | -low sensitivity for detecting small lesions in the duodenum, stomach, and small intestine | -liver, pancreas, brain, and bone evaluation |

**Table 3.** *Cont.*

|  | Advantages | Disadvantages | Use |
|---|---|---|---|
| Fluorodeoxyglucose PET | -complementary information distinguishing between slowly proliferating and aggressive tumors (higher glucose metabolism in G3 and high G2 NENs) | -ionizing radiation<br>-low sensitivity for neuroendocrine tumors | -optional in NEN evaluation<br>-prognostication (worse prognosis if FDG uptake) and post-therapy assessment |
| 68Ga-DOTA-SSA PET/CT | -good sensitivity and specificity | -low sensitivity for insulinomas | -staging with primary tumor location and metastatic detection<br>-restaging with the assessment of residual, recurrent, or progressive disease<br>-patient selection for peptide receptor radionuclide therapy |

CT: computed tomography; MRI: magnetic resonance imaging; PET: positron emission tomography; US: ultrasound; EUS: endoscopic ultrasound; SSA: somatostatin analog.

### 2.1. Functional Assessment

Regarding lesion detection, 68Ga/64Cu-DOTA-somatostatin analogue (SSA) positron emission tomography (PET), in combination with CT (PET-CT), has the higher sensitivity for the detection of most types of NENs, representing, according to the ENETS guidelines, the diagnostic tool of choice to localize the disease in noninsulinoma pancreatic NETs. In a small portion of insulinoma patients (<5–10%), the overexpresses of glucagon-like peptide-1 (GLP-1) can be a target for scintigraphy with radiolabeled GLP-1 receptor analogues [31]. However, this method is confined to research settings. SSTR scintigraphy (SRS) should be carried out only if PET-CT is not available, since this method has a lower sensitivity [25].

According to the European Society of Medical Oncology (ESMO) guidelines [32], the use of PET with [18F] fluorodeoxyglucose (FDG) is optional in NEN evaluation. However, FDG is the tracer of choice for G3 and high G2 NENs, since these entities show a higher glucose metabolism and less SSTR expression compared to low-grade NENs. The presence of FDG uptake in NEN patients at PET-CT suggests a worse prognosis.

### 2.2. Functional Tools

#### 2.2.1. Somatostatin Receptor Imaging (SRI)

SRI can be obtained through scintigraphy, using a gamma camera or, more recently, PET. It is mandatory to perform SRI to obtain information on the status of the somatostatin receptor in lesions to assess the patient's suitability for treatment with somatostatin analogues. Somatostatin is a regulatory peptide widely distributed in the human body, and its action is mediated by membrane receptors, of which five subclasses (sst1-sst5) have been cloned [32]. The subtype predominantly expressed by GEP-NENs is sst2, and its cellular expression decreases with an increasing proliferation (Ki-67) and, consequently, tumor uptake on somatostatin receptor imaging [33,34]. For 18FDG-PET/CT metabolic imaging, the situation is speculatively opposite, with lesions usually becoming more 18FDG avid as proliferation increases. For the detection of higher-grade NETs, 18FDG-PET/CT may, therefore, be preferred. However, studies show that using both somatostatin receptor imaging and 18FDG-PET/CT in a complementary manner increases sensitivity [34,35].

Somatostatin receptor scintigraphy (SRS) using 111In-DTPA-octreotide (OctreoScan) in its original form provides whole-body planar images, while, in modern practice, this image is merged with single-photon emission computed tomography (SPECT) and CT. Diagnostic accuracy is, thus, improved by exploiting the specificity of OctreoScan and the anatomical details provided with SPECT/CT [36]. 111In-Pentetreotide, acquired as SPECT/CT, can be used for the routine diagnostic imaging of nonfunctional NEN-GEPs, with a significant impact on treatment planning, as it exhibits higher sensitivity and accuracy values than

conventional imaging, with an added value of 35.6%. It was also shown to change patient classification and management in 27.9% of cases, reducing disease in 9.6% of cases compared to conventional imaging [37].

The newest somatostatin receptor-based imaging technique employs the positron emitter 68Ga to label various somatostatin analogues [38]. The 68Ga positron emitter has been used to label somatostatin analogues for PET. Two different preparations of octreotide, 68Ga-DOTATOC(DOTA,1-Nal(3)-octreotide), and 68Ga-DOTANOC(DOTA,1-Nal(3)-octreotide), and one of octreotate, 68Ga-DOTATATE(DOTA, and Tyr(3)-octreotate) are the most commonly used [39–42]. The preparations have different affinities with various somatostatin receptor (sst) subtypes. However, the differences between the three preparations seem marginal for the routine clinical use of PET [43]. The main uses of 68Ga DOTA conjugated peptide-binding SSTR are in staging with primary tumor location and metastatic detection, in restaging with the assessment of residual, recurrent, or progressive disease, and, finally, in patient selection for PRRT [37]. PET with 68Ga-labeled somatostatin analogues has been shown in numerous studies to be superior to SRS [32,44–47] and CT/MRI [23–28], although similar results were found in a study comparing 68Ga-DOTANOC and SRS [48]. PET with 68Ga-labeled somatostatin analogues also showed better results than 18F-DOPA [49,50]. The spatial resolution is higher in PET versus SRS (0.5 cm versus 1.5 cm), and tissue contrast is better. Furthermore, PET offers logistical advantages, because the more favorable kinetics of 68Ga-labeled preparations allows for images to be obtained as early as 30–60 min after injection. Finally, the sensitivity reported in various studies of PET/CT ranged from 86% to 100%, and the specificity ranged from 79% to 100% [50–58] for all panNETs, except for insulinomas, in which case, the sensitivity was only 25% [59]. Accordingly, current major guidelines consider 68Ga-labeled somatostatin analogs PET/CT to be the first-line diagnostic procedure for the staging or restaging of any noninsulinoma panNET cases for detecting an unknown primary tumor site or early recurrence, and for the evaluation of patients undergoing radiometabolic peptide receptor therapy (PRRT) [25,60,61].

### 2.2.2. Positron Emission Tomography (PET)

In addition to 68Ga-labeled somatostatin analogues, other PET tracers exist for NEN imaging. NENs were previously described as APUDOMas, and specific PET tracers were developed based on these properties. The most commonly used radiotracer in PET imaging is 18F-FDG, which evaluates the glucose metabolism. It showed imaging results superior to that of octreotate. However, this method is generally considered when other imaging techniques fail or demonstrate contradictory results or limited equivocation. It has been suggested that it may provide complementary information for distinguishing between slowly proliferating and aggressive tumors [62,63]. Additionally, assessing the glucose metabolism with 18F-FDG may allow for a prognostic prediction, including overall and progression-free survival. It may also be useful in the diagnosis of aggressive disease, particularly in high-grade tumors, with better prognostic values and greater sensitivity in identifying the extent of the disease. As in all 18F-FDG-avid tumors, the analogue of glucose accumulates only in high-grade neuroendocrine neoplasms, and, therefore, FDG-PET has not been considered useful for imaging in most patients with NENs [64–66]. In more recent studies, however, FDG-PET has been shown to provide additional information when combined with 68Ga-DOTATATE [67,68] and SRS [34], also providing helpful information in the preoperative diagnostic workup regarding the behavior and aggressiveness of the tumor [69]. It has also been reported that FDG may positivity predict early tumor progression [70] and be associated with a higher risk of death [71]. In a recent study on pancreatic NENs, FDG-PET was used to define tumors with a higher malignant potential [72].

### 2.3. Morphologic Imaging and Tools

Although the limitations of morphological imaging are well known in the management of patients with NENs, the optimization of traditional techniques, technological improvements, and the introduction of new functional sequences in magnetic resonance imaging have achieved high sensitivity in detecting these entities. Furthermore, morphological imaging is essential in local staging to identify patients suitable for surgical resection, since this approach is considered, according to the ESMO guidelines, the first-line treatment for several GEP-NENs.

### 2.3.1. Computed Tomography

CT is usually the initial imaging technique in the oncology field, where the CT examination is often repeated in the course of treatments to conduct a response assessment due to its good sensitivity, specificity, and availability [73–76]. Optimizing techniques are mandatory for assessing GEP-NENs, requiring multiphase CT protocols with a contrast medium [77]. Scans should be performed before the injection (useful for discriminating calcifications), in the arterial phase (typical NEN enhancement and mandatory to increase the detection of small functioning pancreatic NENs, so as to assess the encasement of the hepatic, splenic, and mesenteric artery), and the portal venous phase (for liver imaging and the delineation of parenchymatous organs, as for the encasement of mesenteric and portal veins) [78]. For pancreatic lesions, the contrast study protocol should include the delayed phase for the detection of delayed enhancement presented by some fibrous tumors. CT is an excellent method for detecting complications secondary to GEP-NETs, such as a bowel obstruction, intussusception, or desmoplastic reaction, which are life-threatening diseases [79–87]. The right timing of the arterial phase is crucial for appropriate imaging. NENs have a strictly arterial blood supply and begin taking the contrast medium as soon as it reaches the lesion through the arterial system. With regard to pancreatic lesions, in order to obtain an optimal contrast between the lesion and parenchyma, a flow rate of 4 mL/s has been suggested, with bolus triggering on the descending aorta, and a delay between the trigger and arterial scan of 12 s. When focusing on liver imaging, a slightly later time may be chosen for a more intense impregnation of the lesions. Early scanning after the initiation of a contrast injection (first-stage arterial or CT angiography) may occasionally be required prior to surgery to show arterial anatomy. With postprocessing, the vascular anatomy can also be shown in three-dimensional images, MIP (maximum intensity projection), and using the volume rendering technique. In the late arterial phase (or portal inflow phase), after approximately 10–15 s, the small arterial branches are enhanced and this is the best phase to delineate hypervascular liver metastases and pancreatic NENs. In the venous (or portal venous) phase, approximately 60–90 s after the start of the injection, the contrast medium passes through the capillaries and reaches the veins, including the portal vein, with improved visualization of normal hepatic parenchyma. In this phase, hypovascular liver metastases are better visualized. Since current CT scanners produce images of 1 mm or less, coronal and sagittal two-dimensional images can be obtained with high resolution through multiplanar reconstructions (MPRs) [25,88–90] (Table 4). Such reconstructions can be helpful, for example, to show the relationship between a pancreatic NEN and the pancreatic duct. The use of dual-energy computed tomography (DECT) in diagnosis and follow-up in patients with NETs is growing. Thanks to the use of different X-ray spectra, e.g., 80 and 140 kV, DECT can differentiate tissue samples simultaneously, providing iodine density quantification on a single contrast-enhanced scan without increasing the patient's radiation dose and, as a result, produces images with greater contrast between the lesion and surrounding tissue [91–96]. Noah et al. showed that CT iodine maps could increase radiologists' confidence in studying these tumors [92]. DECT has shown a higher sensitivity detecting pancreatic insulinomas compared to conventional CT (95.7 vs. 68.8%) [97]. According to the ENETS guidelines for jejunum and ileum NENs, CT enteroclysis is the diagnostic tool that should be chosen for small bowel lesions if CT is inconclusive. CT enteroclysis showed a sensitivity of approximately 92%, and a positive predictive value

(PPV) of 95% [98–100]. The patient's preparation is critical for the CT enteroclysis, and involves the use of a neutral contrast medium (water and methylcellulose) with a nasojejunal conduit and the administration of spasmolytics [25]. The contrast study includes an arterial phase (approximately 25 s after the start of the intravenous contrast injection) and a venous phase (60 s after). In addition, extraluminal findings may be assessed as metastases in solid organs or in lymph nodes as a desmoplastic reaction. When the neutral contrast is administrated through drinking approximately 1L of oral negative contrast, we could perform a CT enterography that shows a sensitivity similar to CT enteroclysis [101,102] (Figure 1).

**Table 4.** CT protocols for diagnosis and staging of GEP-NENs.

| Abdomen Multidetector CT Protocol | |
|---|---|
| Voltage | 120 kVp |
| Effective amperage | 200 mAs |
| Rotation time | 0.5 s |
| Detector collimation | 1.5 mm |
| Section thickness | 3.0 mm |
| Pitch | 0.75 |
| Increment | 1.5 mm |
| Coverage | Image from the 11th vertebral body through the iliac crest |
| Oral contrast material | Negative oral contrast material (500 mL of water 30 min before examination and 250 mL of water immediately before examination) |
| Nonenhanced CT | |
| Contrast-enhanced CT | 100–125 mL isomolar or osmolar iodinated contrast material (370 mg/mL) at 4–5 mL/s |
| **Image Acquisition Phase** | |
| Arterial phase | 15–25 s (CT angiography in preoperative setting) 25–30 s |
| Pancreatic parenchymal phase | 40–45 s |
| Portal venous phase | 60–70 s |
| Multiplanar reformation | Axial, sagittal, and coronal planes; section thickness, 3 mm |
| **Thorax/Neck Multidetector CT Protocol** | |
| Thorax–Neck–Abdomen | Amount of contrast media and injection rates adjusted to what is required to perform a proper CT of the abdomen |
| Neck | 1.5–2 mL/kg body weight of contrast media 300–350 mg/mL injected at 2.5 mL/s using a 40 s scanning delay |
| Thorax | 1–1.5 mL/kg body weight of contrast media 300–350 mg/mL injected at 1.5 mL/s using a 60 s scanning delay |
| **CT Enteroclysis** | |
| Patient preparation | 2000 mL of hyperosmolar fluid, such as mannitol, with a 50% concentration, or, alternatively, warm tap water administered through a nasogastrojejunal tube by using a dedicated (150–200 mL/min rate) power injector Intravenous glucagon or anticholinergic drug (recommended) |
| Contrast-enhanced CT | 120–150 mL isomolar or osmolar iodinated contrast media at 3 mL/s Arterial phase using a 25 s scanning delay Venous phase using a 60 s scanning delay |

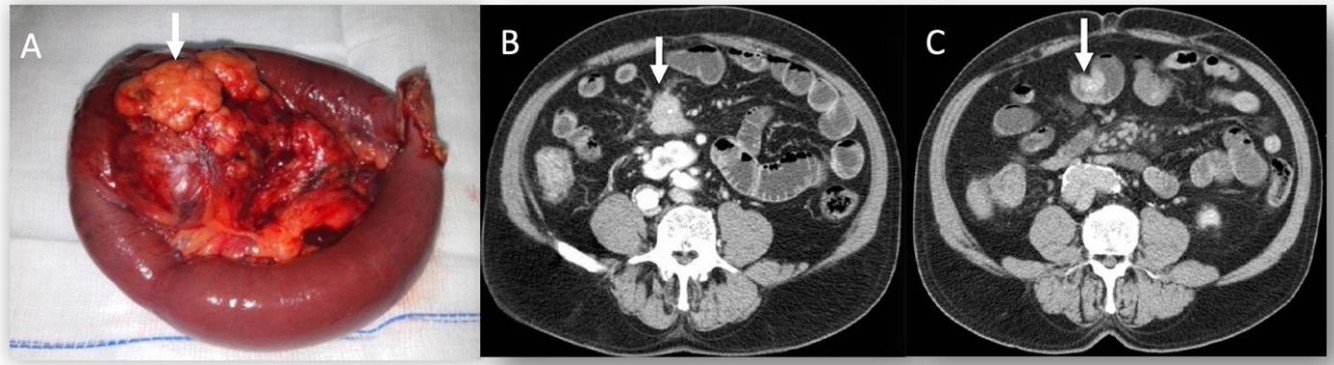

**Figure 1.** Sample size (**A**) of small bowel NEN with desmoplastic reaction (arrows in (**A**) and (**B**)), evaluated on CT (**B**) in portal phase of contrast study. In (**C**), arrow shows the intraluminal small tumor as an enhanced polypoid lesion.

### 2.3.2. Magnetic Resonance Imaging

According to the ESMO guidelines, MRI is the diagnostic tool of choice for liver and pancreas assessments due to its accuracy in the differential diagnosis between primary hepatic malignancy, liver metastases, and benign lesions, such as hemangiomas [103]. However, during the presurgical phase, given the better accuracy of CT in vascular evaluations, the two methods should be complementary. MRI also has a higher accuracy than CT for bone and brain evaluations [104–106]. Otherwise, CT is preferred for the lungs. The MRI sensitivity for PNETs detection is 79% (range 54%-100%) [107–109], while the mean sensitivity of MRI for the detection of liver metastases is 91% (range 82%-98%) [110–114]. In 2015, Flechsig et al. demonstrated the greater sensitivity of MRI in the search for liver metastases, comparing it with both CT and PET with 68Ga-DOTATOC. MRI, where PET fails, would seem capable of detecting metastases thanks to DWI, although a differential diagnosis with focal nodular hyperplasia (FNH) can sometimes be challenging. The sensitivity obtained with DWI and T2-weighted images would seem comparable to those obtained with contrast, which, therefore, can be used to complete the investigation [115,116]. The standard abdominal study protocol can be correlated to the target.

Regarding pancreatic lesions, it generally includes T1- and T2-W conventional sequences, T2-W fat-suppressed (FS) sequences, DWI, cholangiopancreatography sequences, and pre and postcontrast T1-W sequences. Conventional sequences allow for the identification of a lesion's structural characteristics (solid vs. cystic). T2-W FS and DWI increase the detection rate and allow for a better characterization of small lesions, allowing to identify multiple lesions in familial syndromes. Cholangiopancreatography sequences assess the relationship between the lesion and the pancreatic duct for correct characterization and better surgical management. Regarding postcontrast sequences, for pancreatic lesions, due to the retroperitoneal side, it is possible to consider studying these protocols: contrast-enhanced (CE) MRI and dynamic contrast enhancement (DCE) MRI. In CE-MRI, T1-W FS sequences are acquired in specific phases (e.g., arterial, pancreatic, and venous phases), while, in DCE-MRI, more sequences are acquired continuously, without pause, for approximately 5 min. This method allows for a qualitative, semiquantitative, and quantitative evaluation. Usually, CE-MRI is the standard protocol, while DCE-MRI is preferred in research settings.

Regarding liver metastasis assessments, the standard protocol includes T1- and T2-W conventional sequences, T1-W in-phase and opposed phase, T2-W FS sequence, DWI, and postcontrast T1-W sequences. Similarly to the above, conventional sequences allow for identifying a lesion's structural characteristics. T2-W FS and DWI increase the detection rate of small lesions and allow for better characterization. Regarding contrast protocol studies, two subtypes of contrast agents can be employed: a nonspecific contrast medium that distributes into the vascular and extravascular extracellular spaces, and liver-specific agents, taken up by liver cells. The advantage of the liver-specific agent is correlated with the possibility of increasing the detection rate of small liver metastases, considering the pharmacokinetics of these media; during the hepatobiliary phase, normal liver parenchyma is uniformly hyperintense, while primitive or secondary lesions are not hyperintense due to missing normal hepatocytes [117,118]. Despite the proven advantages of hepatospecific contrast agents, recent studies showed that a suboptimal image quality is frequently observed in arterial phase imaging with Gd-EOB-DTP, which could negatively affect the characterization of hepatic lesions. Consequently, these agents should be chosen in presurgical settings, while, during characterization, we must carefully consider the pros and cons [119–125].

Regarding small bowel lesions, MR enteroclysis provides a better assessment of mucosal lesions compared to MR enterography [126]. The study protocol should include, in axial and coronal planes, steady-state precession sequences, since these have a lower sensitivity to motion artifacts. These sequences allow for a rapid overview of the entire abdomen. In addition, coronal and axial T2-W, fast or turbo spin echo sequences, based on the half-Fourier reconstruction technique, should be performed to obtain high contrast between the lumen and the bowel wall. However, these sequences are susceptible to intraluminal motion due to flow void, so a radiologist should have familiarity with these artifacts.

A precontrast and postcontrast coronal T1-W FS gradient-echo sequence are applied during the arterial and the venous phase of the contrast study, followed by similar axial sequences covering the entire abdomen. The addition of DWI has been suggested, with at least two b values. Although, currently, CT enteroclysis is the primary imaging modality in small bowel tumors, few data are available on the comparison of the effectiveness of CT and MR enterography techniques.

Regarding rectal lesions, the study protocol includes T2-W sequences and DWI, while T1-W contrast sequences are optional. Specific parameters should be followed to achieve optimal high-resolution sequences, including a small FOV, slice thickness (no more than 3 mm), and correct scan plane alignment (perpendicular to the rectal wall at the level of the tumor).

Concerning bone metastases, commonly used protocols include a combination of T1-W spin echo and fat-suppressed short-tau inversion recovery (STIR) sequences. Recently, obtaining data in whole-body DWI studies has been attracting significant interest, although most studies are aimed at prostate metastases. According to a neuro-oncology study group [127], in brain metastases, the "minimum standard" recommended pulse sequences include: (i) parameter matched pre- and postcontrast inversion recovery (IR)-prepared isotropic 3D T1-weighted gradient echo (IR-GRE); (ii) axial 2D T2-weighted turbo spin echo acquired after the injection of a gadolinium-based contrast agent and before postcontrast 3D T1-weighted images; (iii) axial 2D or 3D T2-weighted fluid-attenuated inversion recovery; (iv) axial 2D, three-directional diffusion-weighted images; and (v) postcontrast 2D T1-weighted spin echo images for increased lesion conspicuity. Recommended sequence parameters were provided for both 1.5T and 3T MR systems. An "ideal" protocol was also provided, replacing IR-GRE with 3D TSE T1-weighted imaging pre- and post-gadolinium, and is best performed at 3T, for which a dynamic susceptibility contrast perfusion is included.

Regardless of the target, to improve the temporal resolution (number of phases) and avoid artefacts, acquisition acceleration techniques and motion compression techniques are commonly used. Multiphasic imaging and apnea acquisition can be used, but the time between the contrast injection and acquisition is critical following the times described in CT [128]. Furthermore, if we are looking for a PNEN, the use of thin-layer acquisitions is recommended [129]. The use of diffusion also involves the use of respiratory gating or breath compensation techniques with parallel imaging and a b-value above 500, but also the acquisition of at least two b-values or more, for example, (0, 50, 600) [128,130], since it is, thus, possible to calculate ADC (apparent diffusion coefficient) maps [115,131]. The literature on this subject is still limited, but the increased ADC values seem to be linked to those of Ki-67 and also to the evaluation of responses to TACE (transarterial chemoembolization) [131].

Conventional DWI is based on a monoexponential model and assesses the water molecule motion according to a Gaussian approach [132–135].

However, it is known that water molecules diffusion within tissue follows a non-Gaussian model; thus, a non-Gaussian approach, named diffusion kurtosis imaging (DKI), was proposed by Jensen et al. [136].

Using the DKI, it is possible to calculate the kurtosis median coefficient (MK) and to assess the variation of diffusion behavior with a Gaussian to a non-Gaussian approach and the diffusion coefficient (MD), which evaluates the correction of the non-Gaussian bias. DKI allows us to obtain more data on tissue structures than DWI does. However, DKI requires high-quality images at *b*-values greater than 1000 s/mm$^2$ and a high signal-to-noise ratio (SNR) [137].

Shi et al. [107] assessed DKI performance in differentiating pancreatic ductal adenocarcinomas (PDACs) from solid pseudo papillary neoplasms (SPNs) and PNETs, showing that the accuracy rate with DKI was higher than that of a subjective diagnosis alone [138]. In addition, Le Bihan et al. [139,140] proposed a biexponential model, the intravoxel incoherent motion (IVIM), to obtain data on tissue perfusion. By using the IVIM model and multiple, sufficiently low *b* values (<200 mm$^2$/s), not only can pure diffusion characteristics (*D*) be separated from pseudodiffusion caused by microscopic circulation in tissue, but perfusion characteristics (pseudodiffusion coefficient (*D**)) and their proportions (perfusion fraction (*f*)) can also be extracted.

Zeng et al. showed that conventional DWI and IVIM models are valuable tools to differentiate nonhypervascular PNETs from PDACs. *D** showed better performance than *f* and ADC [141]. Improvements in scanner performance and the optimization of pulse sequences have reduced acquisition times and increased the use of whole-body MRI (WB-MRI) in several clinical scenarios [142]. Few studies report [58,143] on the use of WB-MRI and PET in PET/MRI hybrid scanners for NET imaging [144,145]. In a total study time of 1h or less, the exam protocols include the neck–thorax–abdomen (and brain when needed), with the acquisition of DWI and IV contrast-enhanced images of the liver and pancreas. Finally, although MRI is used less frequently than US or CT as a guide for minimally invasive interventional procedures [146–158], its use has increased. It has many advantages, such as a lack of ionizing radiation, real-time MR fluoroscopy placement, high resolution, capacity to display small tumors with increased sensitivity, and the monitoring of thermal effects. It can be combined with diffusion-weighted imaging or MRI contrast agents to visualize more difficult lesions [159,160] (Figure 2).

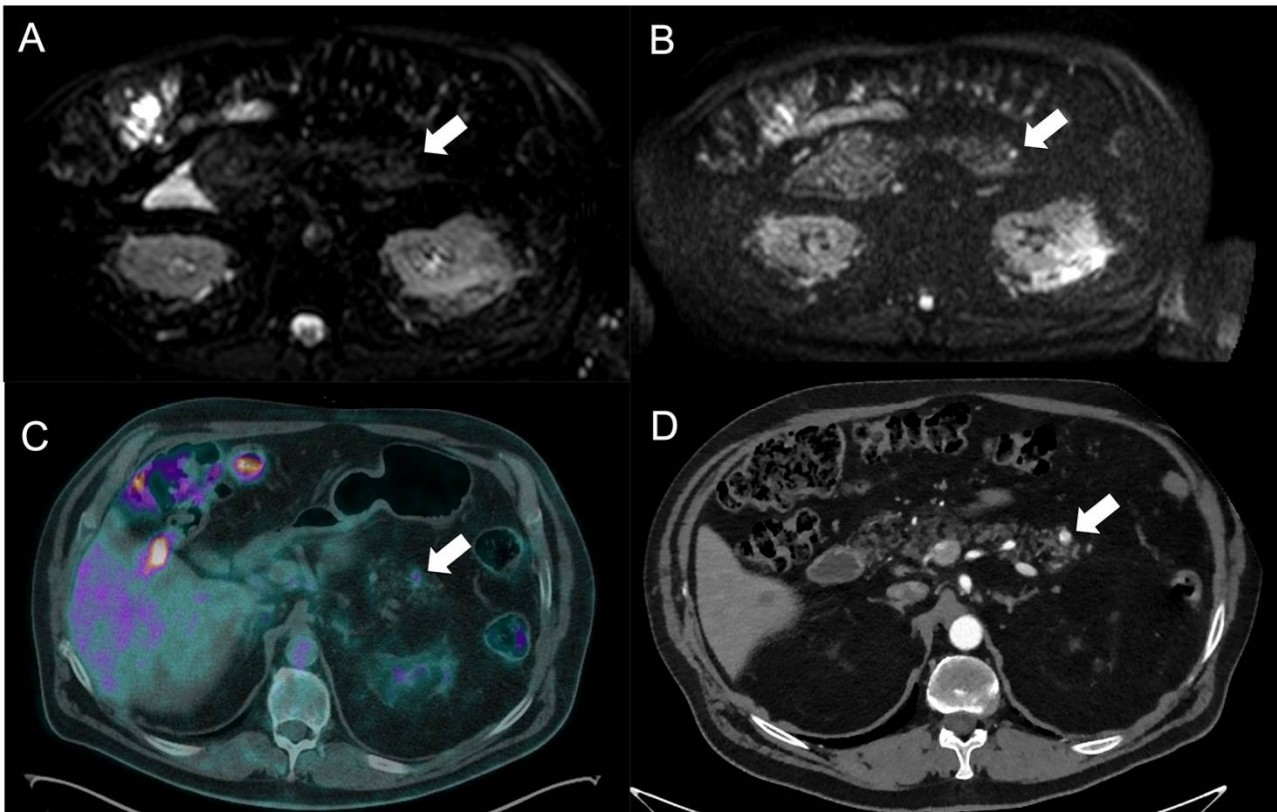

**Figure 2.** The pancreatic NEN lesion at the tail is highlighted with an arrow. DWI at low and high b-values, respectively, in Subfigures (**A**,**B**), showing the restricted diffusion detecting the little lesion, typically characterized by 68Gallium uptaking in PET/CT (**C**) and high arterious enhancement at CT (**D**).

### 2.3.3. Ultrasonography and Contrast-Enhanced Ultrasound (CEUS)

Although the abdominal US is a widely used, easy-to-perform, and inexpensive diagnostic tool [161], its role in GEP-NEN evaluations is limited and, in some cases, controversial.

According to the Polish Network of Neuroendocrine Tumours, abdominal US is an excellent tool in liver metastasis assessments with a sensitivity of 85–90%, and the possibility to use a contrast medium (CEUS) increases the sensibility and allows to characterize liver lesions that remain equivocal on CT/MRI [162]. Additionally, although US is not a technical standard for therapy monitoring, it may be helpful in surveillance in some patients with liver metastases not clearly visible on CT and MRI or in young patients, to reduce the radiation dose. The differences in sensitivity are due to anatomical conditions, cooperation with the patient and the experience of the physician [162]. In addition, it may be challenging in patients with a large tumor burden to obtain an overview of an enlarged liver in which most of the normal parenchyma is replaced with the tumor. Of contrast, abdominal US should not be proposed in pancreatic lesions since its sensitivity is of 13% to 27%. In comparison, endoscopic ultrasound (EUS) is the current optimal diagnostic tool to detect small PNETs with 86% (range 82%-93%) sensitivity and 92% (range 86%-95%) specificity [163]. EUS plays a decisive role in defining GEP-NENs of the gastrointestinal wall, as it provides information on the size, depth of invasion, and local–regional metastases [164]. The EUS-guided biopsy can also provide a definitive diagnosis and useful information (for example, the evaluation of the nuclear protein Ki-67, which appears to be a proliferative index) for correctly managing this type of lesion [165]. In addition, the EUS can accurately select the ideal candidates for the endoscopic resection. EUS plays a decisive role in the setting of pancreatic NENs. It can help locate the tumor correctly when other noninvasive procedures have failed and provide useful additional information (e.g., the distance to the

pancreatic duct and the Ki-67) for the best therapeutic management (surgery, conservative approach, and type of anticancer therapy in cases of unresectable tumors). EUS is an essential method for the detection of small PNENs [166]. In addition, contrast-enhanced endoscopic ultrasound using second-generation contrast agents allows for the assessment of pancreatic tumor intratumoral vascularity by permitting the differentiation of pancreatic ductal adenocarcinoma from PNENs, with the latter generally showing a hypoenhancement pattern, while PNENs typically show a pattern of hyperenhancement [167]. Indeed, it has been shown that CT fails to find 68.4% of PNENs <10 mm and 15% of those ≤20 mm in diameter [168]. In a meta-analysis, James et al. reported that preoperative EUS increases the overall PNEN detection rate by >25% after a CT scan, with or without additional investigative examinations, such as MRI or ultrasound [169]. Additionally, the ability to perform EUS-guided tattooing in pancreatic lesions can help surgeons find cancer and avoid primarily destructive surgeries. Finally, EUS-guided therapies (e.g., alcohol ablation), especially in patients unsuitable for surgery, are under investigation, and could represent a future field of interest [131]. The transducer is placed directly on the organ's surface in perioperative ultrasound (IOUS) and is mandatory in the surgical resection of a pancreatic NEN to facilitate the diagnosis of liver metastases during surgery. IOUS facilitates lesion detection and localization in the pancreas and liver, and, according to the ESMO guidelines, it is mandatory before pancreatic resection in MEN1 syndrome patients [170].

Generally, on US images, panNENs present as well-defined, solid, and heterogeneous hypoechoic lesions, and some of them can present with cystic regions [171,172]. Liver metastases often appear hyperechoic compared to the surrounding liver parenchyma, although they can also manifest with a hypoechoic and targetoid appearance with increased vascularity on Doppler US (Figure 3).

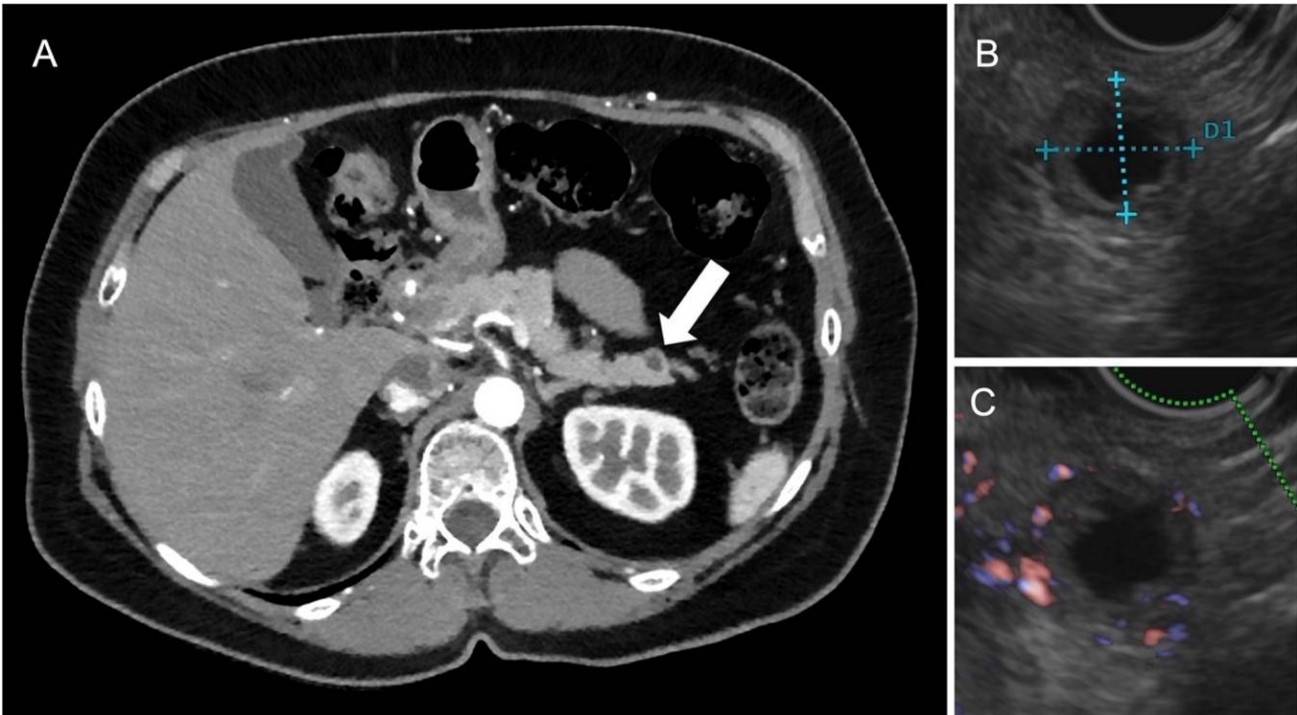

**Figure 3.** Arterious CT scan (**A**) demonstrating a hypodense lesion at pancreatic tail (arrow), which, at echoendoscopy, showed anechoic content (**B**) with hypervacularized thick walls at Doppler, and after a bolus of ultrasound contrast agent (**C**). The tumor was surgically removed and diagnosed as G1 NEN tumor with cystic appearance.

## 3. Clinical Setting

### 3.1. Pancreatic NENs

According to the ESMO guidelines, surgical resection is the treatment of choice in G1 and G2 NENs. In functional lesions, clinical symptoms should be managed before any approach.

Regarding PNENs, several studies showed the safety of a watch-and-wait approach instead of surgical resection for asymptomatic NF-PNETs <2 cm [60,173,174].

Regardless, surgical resection is the front-line treatment in young patients, in cases of local invasiveness, and in the presence of functioning PNETs, irrespective of tumor size. A parenchyma-sparing resection (e.g., enucleation or central pancreatectomy) should be considered in cases of the absence of signs of invasiveness, as the dilation of the main pancreatic duct and nodal involvement. If there is nodal involvement, a standard pancreatectomy with lymphadenectomy is mandatory [170].

In this scenario, imaging may identify single or multiple lesions and define the localization, size, functional activity, and signs of local invasiveness. It, therefore, appears clear that there is not one technique better than another, but there is the necessity of multimodal imaging, where the morphological ones and vice versa support the functioning techniques. Indeed, while functional techniques allow to establish the presence or absence of one or more lesions and the functional status, MRI efficiently identifies the localization, the relationship with the main duct and the presence of liver metastases [112,175,176]. CT allows a better vascular assessment even in the presence of anatomical variants, so as pulmonary metastases [177,178] (Figure 4).

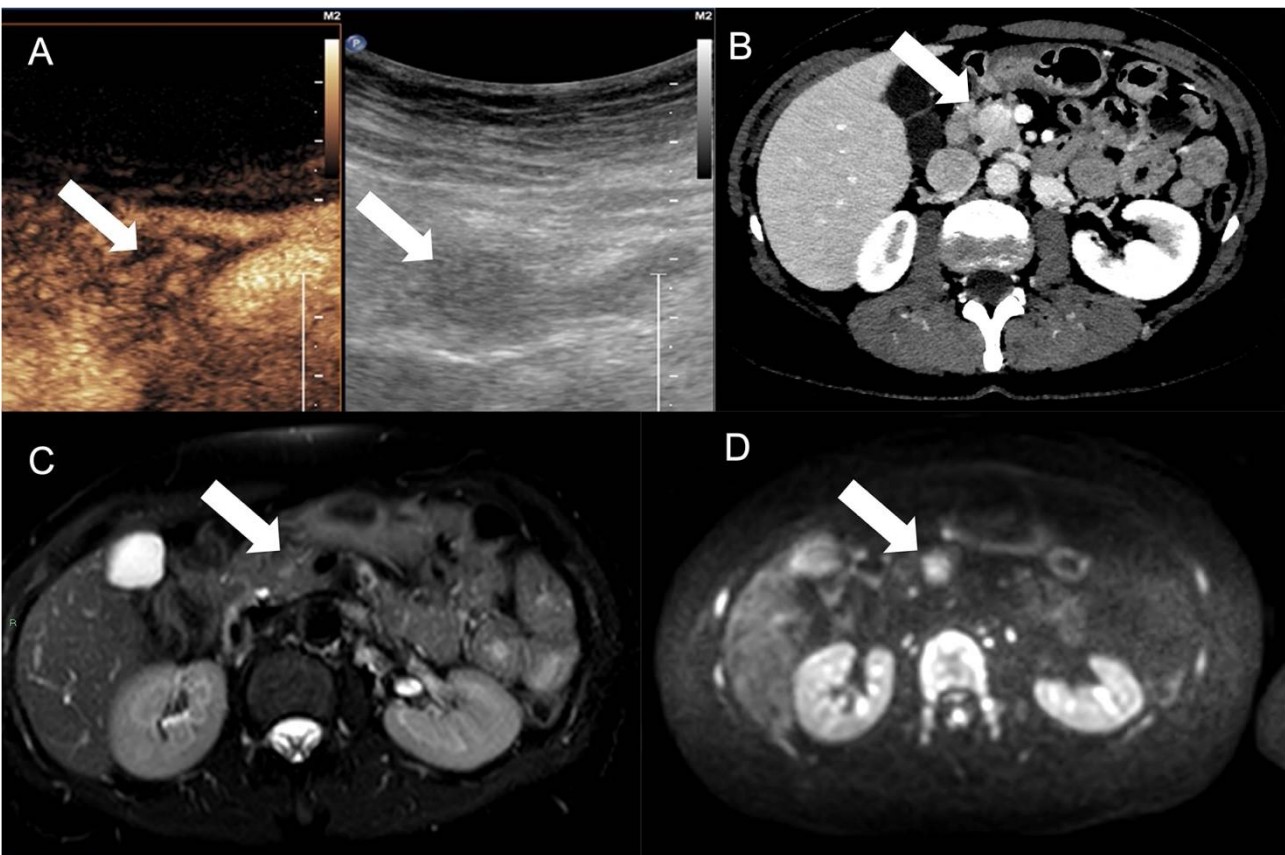

**Figure 4.** A pancreatic NEN tumor (arrow) was discovered as hypoechoic mass at ultrasound, showing high vascularization at CEUS (**A**) and CT (**B**). At MRI examination, the mass was not clearly defined with T2W fat-suppression sequence (**C**), while the restriction at diffusion imaging improved the definition of the mass (**D**).

*3.2. Liver Metastases*

Surgery is a recommended treatment option for most well-differentiated G1 and G2 tumor neuroendocrine liver metastases (NELMs) in unilobar and limited diseases [14]. The surgical approach is influenced, in fact, by lesion size and location, disease stage, and the patient's symptoms [179]. Due to the frequent bilobar and multifocal localization, less than 20–30% of patients are candidates for liver resection with curative intent. Liver transplantation as a therapeutic option is feasible in selected patients with unresectable NELMs [180–183]. Moreover, cytoreductive ablative therapies, in addition to surgical resection, can offer improved survival and quality of life at 5 years compared to patients who do not undergo surgery (70%–90% vs. 50%) [184]. For patients with NELMs who are not candidates for surgery for high liver involvement or sites that are inaccessible or refractory to medical therapy, intra-arterial treatments, such as transarterial embolization (TAE), transarterial chemoembolization (TACE), and selective internal radiation therapy (SIRT), are safe and effective options for gaining control of the disease. These techniques allow for biological and symptom improvements in selected patients, or are palliative treatments in patients with multiple, bilateral, or unresectable NELMs, reducing tumor volume and endocrine secretion [185]. Intra-arterial treatments are effective because NELMs are usually hypervascular tumors with a predominant arterial supply. Intra-arterial therapies in NELMs have demonstrated efficacy in a subset of NELMs (i.e., those from midgut origin) [186], while they must be associated with intra-arterial chemotherapy for liver metastases from colorectal cancer [187]. NELMs may be linked to biological syndromes due to their endocrine secretion. Hence, in addition to the local control of tumor growth, one of the goals of the intra-arterial treatment is to control the endocrine secretion of NELMs, which is not the case for liver metastases of other origins. An improvement in symptoms was observed in 60–90% of patients [188–191]. The mass effect due to liver involvement through NELMs decreased in 100% of patients [192,193]. The therapeutic decision should be discussed in dedicated multidisciplinary meetings [194], and at least one dedicated liver CT or MRI examination should be performed to assess the arterial anatomy, the extent of liver involvement, portal vein patency, and bile duct dilatation. Thoracoabdominal–pelvic CT is indicated for use when looking for extrahepatic metastases from NETs. Ideal candidates have a nonpancreatic NET with a resected primary tumor site, no extrahepatic metastases, and between a 30% and 50% liver involvement. However, some benefits may be observed in other patients, and all patients should be discussed individually. SIRT consists of the transarterial deposition of a radioactive source within tumors [89], based on the predominant arterial vascularization in NELMs [195], and has shown promising results with good tolerance compared to TAE and TACE. Indeed, several studies reported that a complete response occurred in 1–8% of patients, which was not found in TAE/TACE [196–200].

RECIST 1.1 or mRECIST criteria were performed in radiological follow-ups [197,201] after SIRT, and treated lesions appeared hypovascular or necrotic in 97% of patients [202]. Some meta-analyses have shown an objective radiological response rate with a weighted mean of 51% (95% CI: 47–54%) [196,203] and a mean disease control rate of 86–88% [196,204]. Symptom improvement has been reported in 20–80% of SIRT-treated NELM patients and decreased lesion diameters, low enhancement, and a high proportion of necrosis post-SIRT compared to pre-SIRT MRI are prognostic factors of survival after SIRT [205,206]. A decrease in lesion diameters, low enhancement, and a high portion of necrosis on post-SIRT MRI compared to pre-SIRT MRI are prognostic survival factors after SIRT [205,206]. In molecular imaging, a decrease in the standardized uptake value (SUV) ratio of the liver–spleen uptake using 68Ga-DOTATOC PET between pre- and post-SIRT exams has been associated with improved overall survival and progression-free survival [207]. Several retrospective studies suggest that the resection of metastatic lesions improves the 5-year survival rate from 30–40 to 75% [208,209] with a radical resection (R0) of NELMs correlating with a 5-year overall survival (OS) rate of approximately 85% [210]. However, NELMs are frequently more extensive than those identified on presurgical imaging and intraoperatively,

and an accurate curative resection is difficult to achieve. Several pieces of research showed as a multimodality approach with surgical and ablative treatments, allowing for better disease control. So, it is clear the necessity to choose the imaging tool that allows the identification of all lesions so as the relationship with the vascular and biliary duct for proper patient management. Although MRI with liver-specific contrast agents allows to identify very small lesions, this tool should be combined with CT to obtain an optimal vascular evaluation [211–218] (Figure 5).

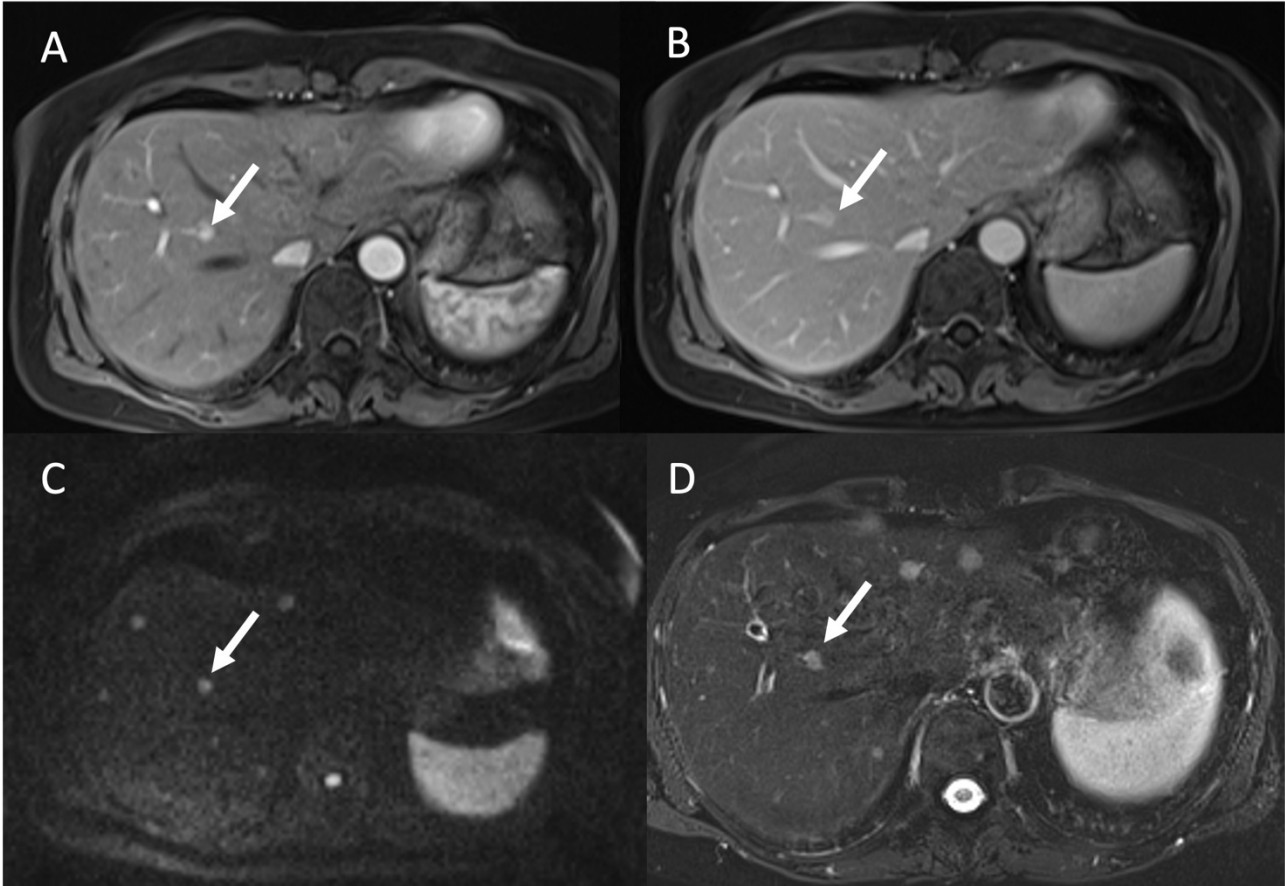

**Figure 5.** MRI evaluation of liver NEN metastases. Contrast study ((**A**): arterial; (**B**): portal phase) show a single lesion (arrow), while, in DWI (**C**) and T2-W FS (**D**), more lesions were detected compared to (**A**,**B**).

*3.3. Gastrointestinal Tract NEN*

In the gastrointestinal tract, neuroendocrine neoplasms are found most frequently in the small bowel (26%) and rectum (34%), less frequently in the appendix (6%), cecum (6%), stomach (12%), and duodenum (8%), with the colon being a rare site of onset [219–223]. Most gastroenteropancreatic neuroendocrine neoplasms are small (<2 cm), and, therefore, conventional imaging techniques fail to define primary cancer in patients with metastatic diseases [7]. For these reasons, other small bowel imaging techniques, such as CT and MRI enterography, or enteroclysis and endoscopic techniques, such as capsule endoscopy and double-balloon enteroscopy, may improve the detection of primary occult tumors. Furthermore, these neoplasms stimulate a fibrotic reaction in the surrounding tissue, which can lead to functional obstruction or vascular compromise [224]. Presently, the use of entero-MRI for diagnosing small intestinal neuroendocrine tumors has not yet been established, considering the high cost of this investigation. The significant advantage of entero-CT compared to conventional CT is the ability to identify lesions of 1–2 cm thanks to the high difference in contrast between neoplasm and intestinal lumen obtained through

the oral administration of the hyperosmolar contrast medium, thus, being able to identify neuroendocrine tumors in a small size also [7]. At the duodenum level, they appear as small intramural masses, polypoid lesions that protrude into the lumen, or circumferential thickenings of the wall. In contrast, when they arise in the small bowel, they can appear as solitary or multiple polypoid lesions presenting an intense enhancement after the intravenous administration of an iodinated contrast medium, or as "plaque" wall thickening [225]. Colorectal GEP-NENs are associated with a worse prognosis when compared with neuroendocrine tumors occurring elsewhere, as they are often advanced at the time of diagnosis. At the colic level, neuroendocrine tumors tend to be larger (>2 cm) and often involve the cecum and ascending colon. Approximately 34% of gastrointestinal neuroendocrine tumors are located in the rectum and often present as small solitary submucosal masses, as multiple nodules, or as single ulcerated polypoid lesions. A virtual colonoscopy cannot differentiate a neuroendocrine tumor from the more frequent adenocarcinoma, since both can appear as circumferential thickening of the wall or as polypoid formations protruding into the intestinal lumen, which are associated, in most cases, with locoregional lymphadenopathies [226] (Figure 6).

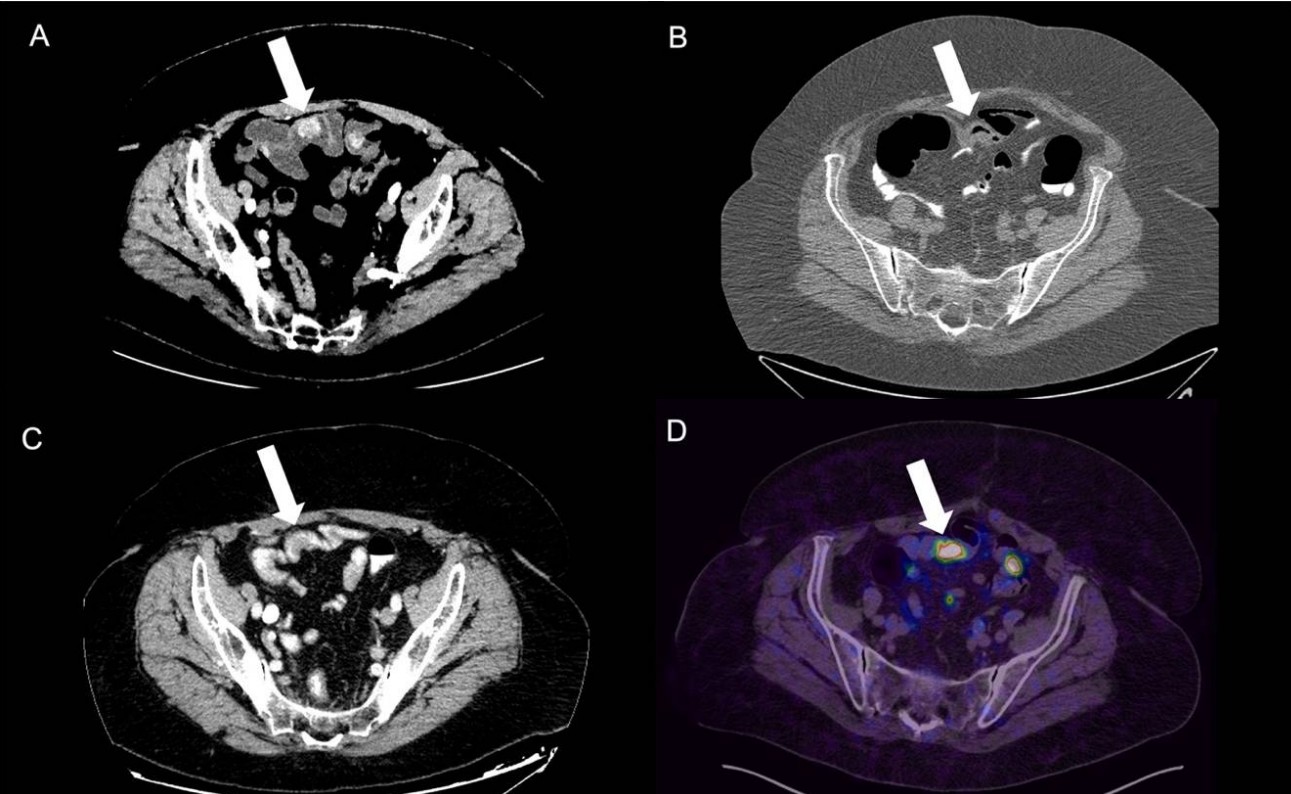

**Figure 6.** Ileal NEN tumors shown with different endoluminal contrasts and in PET-CT exam. The hypervascularized tumors are clearly different from normal bowel walls due to being highlighted in relation to endoluminal hypodense and hyperosmolar contrast media. The same patients previously had a colon CT performed, showing a slight air distension of lumen with minimal wall thickening that did not improve the tumor evidence, as well as a later contrast-enhanced CT with oral hyperosmolar iodinated contrast, which improved the intestinal lumen visualization but strongly limited the enhancement of tumors. PET/CT-68Gallium showed the same highly uptaking lesions, with the highest sensitivity compared to others, but this last one is not routinely performed without neuroendocrine suspicion being specific for highly probable or known NEN tumors. (**A**) CT-enterography, (**B**) virtual colonoscopy, (**C**) CECT with oral contrast, and (**D**) PET-68Ga.

## 4. Follow-Up and Treatment Assessment

There is no consensus on the optimal follow-up for completely resected gastroenteropancreatic neuroendocrine tumors. Published guidelines for follow-up are complex and emphasize closer surveillance in the first 2 years after resection [100,227–229]. The recommendations are summarized in Table 5.

**Table 5.** Follow-up recommendations after NET resection in current guidelines.

| | ESMO | NCCN | ENETS |
|---|---|---|---|
| Small-bowel NEN | Grades 1–2: Laboratory tests and CT or MRI every 3–6 months. Grade 3: Every 2–3 months. Octreoscan after 18–24 months if SRS positive. | Clinical review at 3–12 months with biomarkers and CT or MRI as clinically indicated; then, review every 6–12 months for maximum of 10 years. | Grade 1: US, CT, or MRI at 6 and 12 months, then yearly or longer; octreoscan (or gallium-68-based PET) at baseline and every 2 years. Grade 2–3: US, CT, or MRI every 3 months indefinitely; octreoscan (or gallium-68-based PET) at 3 months and yearly. |
| Pancreatic NEN | Grades 1–2: Laboratory tests and CT or MRI every 3–6 months. Grade 3: Every 2–3 months. Octreoscan after 18–24 months if SSTR positive. | Clinical review at 3–12 months with biomarkers and CT or MRI as clinically indicated; then, review every 6–12 months for maximum of 10 years. | Grade 1: US, CT, or MRI at 6 and 12 months, then yearly or longer; octreoscan (or gallium-68–based PET) at baseline and every 2 years. Grades 2–3: US, CT, or MRI every 3 months indefinitely; octreoscan (or gallium-68-based PET) at 3 months and yearly. |
| Appendiceal NEN | Not reported | Not follow-up: <2 cm completely resected by appendicectomy "as clinically indicated". | No follow-up: <1 cm completely resected with appendicectomy; appendiceal NET >1 cm completely resected with right hemicolectomy without lymph node involvement. |
| Rectal NEN | Not reported | Not follow-up: <1 cm with negative margins. | No follow-up: completely resected rectal NETs <1 cm. |

ENETS, European Neuroendocrine Tumor Society; ESMO, European Society for Medical Oncology; NCCN, National Comprehensive Cancer Network; NET, neuroendocrine tumor; CT, computed tomography; MRI, magnetic resonance imaging; PET, positron emission tomography; SRS, somatostatin receptor scintigraphy; US, ultrasonography.

However, NENs have a different pattern and timescale of recurrence and, thus, require a more practical and tailored follow-up, including improvements in cost, patient flow, and the elimination of excessive demand for inefficient tests [230]. CT and MRI should be preferred for G1 and G2, while the follow-up of G3 tumors should also include gallium-68-based PET. Regardless, the follow-up for NENs requires a multidisciplinary approach, including biochemical (chromogranin A, hormones, and vasoactive amines), radiologic, and histologic investigations.

Regarding treatment assessments, medical therapies in NEN patients vary from somatostatin analogs, targeted agents, such as everolimus and sunitinib, and PRRT to ablation treatments. In this scenario, the limits of Recist 1.1 are clear, which evaluates the tumor shrinkage and the necessity to use functional criteria.

## 5. Conclusions

The radiological approach to GEP-NENs is variable and depends on symptoms, laboratory conditions, and clinical suspicions. In the absence of these, the sensitivity and specificity of instrumental examinations are limited, above all, due to the difficulty in finding primitive lesions, which are often small in size. When symptoms are subtle due to the absence or poor secretion of hormones or neuropeptides, diagnosis is delayed and the prognosis worsens.

The choice of accurate imaging of NETs is fundamental for the diagnosis, staging, and assessment of suitability for surgery, choice of therapy, and response to treatment. The radiologist should be aware of each method's strengths and limitations and their complementarity to recommend the best test for the given clinical scenario. Whichever test is chosen, attention to detail in acquiring and interpreting the study is critical to achieving the best possible results.

**Author Contributions:** Conceptualization, F.P. and V.G.; methodology, M.S.; validation, R.F. and F.G.; formal analysis, V.G.; investigation, L.P., G.T. and S.T.; writing—original draft preparation, F.P.,V.G., L.P.,G.T, R.F., S.T. and F.G.; writing—review and editing, F.P. and V.G.; supervision, M.S. All authors have read and agreed to the published version of the manuscript.

**Funding:** This research received no external funding.

**Institutional Review Board Statement:** Not applicable.

**Informed Consent Statement:** Not applicable.

**Data Availability Statement:** Not applicable.

**Conflicts of Interest:** The authors declare no conflict of interest.

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
