# Peer review of "Diagnostic Management of Gastroenteropancreatic Neuroendocrine Neoplasms: Technique Optimization and Tips and Tricks for Radiologists"

_tomography, doi:10.3390/tomography9010018_

Round 1
Reviewer 1 Report
Point 1: The manuscript presents a comprehensive review of the diagnostic management of gastroenteropancreatic neuroendocrine neoplasms.
Upon reading the manuscript for the first time a number of grammar and English language errors are noted. It is recommended that the manuscript is reviewed again for grammatical and spelling errors.
Point 2: Line 122: The authors state that a histological diagnosis is mandatory in all patients. However, it is well known that it is not possible to always obtain a tissue sample, especially for tumours located in remote areas (for example, the tail of the pancreas). In addition to that, there is a significant number of patients with NEN that do not have access to high volume centres and are treated in minor hospitals without access to advanced interventional services. It would be beneficial to further elaborate on the issue and present imaging options and follow-up when obtaining a tissue sample is not available.
Point 3: Line 228-279: The authors are encouraged to provide additional details on image acquisition and protocols regarding multiphase CT imaging.
Point 4: Additional figures displaying examples of MRI and US examination would be beneficial for readers not accustomed with different imaging modalities.
Point 5: Line 500-518: It is well known that interventional radiology procedures are becoming the mainstay of treating metastatic liver disease, especially for oligometastatic disease, which is often the case with neuroendocrine tumours. The authors should expand this section regarding interventional radiology procedures in NELM
Author Response
We thank the Reviewer for the careful attention dedicated to our manuscript and for all the insightful indications. We believe that our revised manuscript has importantly improved by following the Reviewer’s indications.
Point 1: We thank the Reviewer for this helpful comment which enabled us to correct the inaccuracy in grammatical and spelling errors. Changed phrases are marked using the "Change Track" function.
Point 2: Line 122: We thank the Reviewer for this valuable suggestion. Respecting the indications of the Reviewer, we have modified the text in paragraph 2. Changed phrases are marked using the "Change Track" function.
Point 3: Line 228-279: We thank the Reviewer for this helpful tip. Following the reviewer's instructions, we added Table 4, which summarizes multiphase CT protocols.
Point 4: We are grateful to the reviewer for the chance to add more figures showing examples of MRI and US examinations on pNET. According to his suggestion, we added Figure 4.
Point 5: Line 500-518: We thank the Reviewer for the opportunity to expand on this critical topic. According to his indications, we have expanded this section (section 3.2). Added phrases are marked using the "Change Track" function.
Reviewer 2 Report
I've carefully read the manuscript "Diagnostic management of GEP-NEN: technique optimisation and tips and tricks for radiologist", which is a comprehensive summary of various imaging methods for diagnosis, staging and follow-up of digestive neuroendocrine lesions.
The manuscript is well organised and describes in detail protocols for imaging techniques, benefits and limitations of various techniques. Concerning the therapeutic role of EUS mentioned in the paper, I would suggest adding more data on ablation - with recent data on radiofrequency ablation.
Also, to make the work more attractive, I would suggest adding a figure/image/table summarising the role of each imaging method - CT, MRI, SRI, PET - for diagnosis/staging/follow-up, along with advantages and disadvantages.
Author Response
We thank the Reviewer for the special attention dedicated to our manuscript and for all the appropriate indications. We feel that our revised manuscript has improved following the Reviewer's exhortations.
Point 1:We thank the Reviewer for this valuable advice, allowing us to explore this relevant topic. According to his indications, we have expanded this section (section 3.2). Added phrases are marked using the "Change Track" function.
Point 2: Thanks for the advice, and following this useful indication, we have added Table 3.